# Phantom Study on the Robustness of MR Radiomics Features: Comparing the Applicability of 3D Printed and Biological Phantoms

**DOI:** 10.3390/diagnostics12092196

**Published:** 2022-09-09

**Authors:** Gergő Veres, János Kiss, Norman Félix Vas, Piroska Kallos-Balogh, Nóra Beatrix Máthé, Martin Lyngby Lassen, Ervin Berényi, László Balkay

**Affiliations:** 1Division of Radiology and Imaging Science, Department of Medical Imaging, Faculty of Medicine, University of Debrecen, 4032 Debrecen, Hungary; 2Doctoral School of Neuroscience, Faculty of Medicine, University of Debrecen, 4032 Debrecen, Hungary; 3Doctoral School of Molecular Medicine, Faculty of Medicine, University of Debrecen, 4032 Debrecen, Hungary; 4Division of Nuclear Medicine and Translational Imaging, Department of Medical Imaging, Faculty of Medicine, University of Debrecen, 4032 Debrecen, Hungary; 5Department of Clinical Physiology, Nuclear Medicine and PET and Cluster for Molecular Imaging, Section 4011, Rigshospitalet, University of Copenhagen, 1165 Copenhagen, Denmark

**Keywords:** image processing, texture analysis, magnetic resonance imaging, phantom study, radiomics, 3D printing

## Abstract

The objectives of our study were to (a) evaluate the feasibility of using 3D printed phantoms in magnetic resonance imaging (MR) in assessing the robustness and repeatability of radiomic parameters and (b) to compare the results obtained from the 3D printed phantoms to metrics obtained in biological phantoms. To this end, three different 3D phantoms were printed: a Hilbert cube (5 × 5 × 5 cm^3^) and two cubic quick response (QR) code phantoms (a large phantom (large QR) (5 × 5 × 4 cm^3^) and a small phantom (small QR) (4 × 4 × 3 cm^3^)). All 3D printed and biological phantoms (kiwis, tomatoes, and onions) were scanned thrice on clinical 1.5 T and 3 T MR with 1 mm and 2 mm isotropic resolution. Subsequent analyses included analyses of several radiomics indices (RI), their repeatability and reliability were calculated using the coefficient of variation (CV), the relative percentage difference (RPD), and the interclass coefficient (ICC) parameters. Additionally, the readability of QR codes obtained from the MR images was examined with several mobile phones and algorithms. The best repeatability (CV ≤ 10%) is reported for the acquisition protocols with the highest spatial resolution. In general, the repeatability and reliability of RI were better in data obtained at 1.5 T (CV = 1.9) than at 3 T (CV = 2.11). Furthermore, we report good agreements between results obtained for the 3D phantoms and biological phantoms. Finally, analyses of the read-out rate of the QR code revealed better texture analyses for images with a spatial resolution of 1 mm than 2 mm. In conclusion, 3D printing techniques offer a unique solution to create textures for analyzing the reliability of radiomic data from MR scans.

## 1. Introduction

Conventional analyses of cancerous lesions rely on invasive histological samples, which have several limitations, including discomfort for the patient and potential problems extracting the whole lesion for total-lesion heterogeneity analysis. Therefore, alternatives to the invasive methods are desired. One possible solution is the non-invasive, in-vivo radiomic assessment of images acquired in radiological and nuclear medicine settings [1,2,3,4,5,6]. The major strength in using radiomics as an alternative to histological sampling is the multitude of imaging series, modalities, and follow-up acquisitions employed in the clinical assessment of patients. Radiomic assessments of the various imaging modalities may give more personalized treatment regimes without requiring additional testing of the patients.

However, several challenges have been identified in the clinical translation of radiomics which may affect their applicability in the clinical routine. First, the spatial resolution of the diagnostic imaging modalities (spatial resolutions of 0.5–5 mm) are orders of magnitude worse than the histological samples (spatial resolution of 10^−4^–10^−3^ mm). Second, the choice of radiomic features may be affected by the diagnostic image modality and their modality-specific reconstruction settings [7,8,9,10,11]. Third, image discretization and normalization procedures may affect the radiomic analyses. While these problems have been identified, previous studies have reported great potential in the radiomic assessments for both CT and nuclear medicine imaging settings, suggesting that the discrepancy in spatial resolution nor the choice of imaging modality does not hinder radiomic assessments.

In terms of discretization of the images, two methods exist: the fixed bin size (FBS) and fixed bin number (FBN) [12]. Previous studies have identified that FBS may be the method of choice for quantitative imaging modalities (PET and CT); however, its feasibility on non-quantitative images (MRI) has not yet been evaluated. Therefore, it is of interest to assess how different discretization processes and radiomic features may be employed in MR imaging. Recently, the Image Biomarker Standardization Initiative (IBSI) has sought to standardize the radiomic feature extraction process, resulting in nearly 1000 different radiomics parameters [12]. One way to assess the influence of the discretization processes may include using physical phantoms, such as fruits and vegetables. However, identifying the most suitable phantom in MR imaging is not trivial, as the choice may affect both repeatability and reliability of the radiomics indexes (RIs) during analyses [9,10,13,14,15,16,17,18,19,20,21,22,23,24,25,26,27]. Previous studies have mainly relied on biological phantoms, which decay over time, thus limiting their applicability in longitudinal and multi-center studies [14,15,16]. Recent developments in 3D printing have facilitated the possibility of printing image modality-specific phantoms at a low price and fast production time, which may serve as an alternative to biological phantoms [27,28,29,30,31]. The 3D-printing technique permits manufacturing complicated phantoms, which may arise from mathematical objects or real patient data. These phantoms are strict in shape and consistency and give the possibility to ignore or emphasize even a small number of details. However, dedicated 3D designed and printed MR radiomics phantoms have not yet appeared in the MR imaging literature.

This study aimed to compare radiomics obtained for biological and 3D printed phantoms. To this end, we developed fillable Quick Response (QR) code- and Hilbert curve-based 3D printed models to introduce new MR radiomic phantoms. QR codes and Hilbert curves are well structured and precisely defined mathematical objects, ensuring decent reproducibility in manufacturing [32,33]. QR codes may provide an additional analytical option through the potential of the read-out success rate of the information stored on a deeper level in MR images [34]. The 3D-printed phantoms were compared to biological phantoms that have previously been found useful in the literature (kiwis, onions, and tomatoes) [35].QR codes and Hilbert curves are well structured and precisely defined mathematical objects, ensuring decent reproducibility in manufacturing [32]. QR codes may provide an additional analytical option through the potential of the read-out success rate of the information stored on a deeper level in MR images [34].

## 2. Methods

### 2.1. Biological Phantoms

Given the high water content, vegetables and fruits reflect different signal intensities, shapes, and “tissue” textures in MR images, making them ideal as biological phantoms [13,15,25,36]. For this study, the biological phantoms were selected using the following criteria: the phantom must be water-rich, have an appropriate size (below 5 × 5 × 5 cm^3^), have certain degrees of hardness and heterogeneity, and have stable structural characteristics. Following the selection criteria, three fruits and vegetable types were considered: kiwis (n = 4), tomatoes (n = 3), and onions (n = 3). Representative MR images of one of the selected kiwis are shown in Figure 1. For every acquisition, all the fruits and vegetables were placed in two separate holders to fix and standardize the place of the phantoms during the acquisition steps. Appendix A is shown the placement of biological phantoms. Further, the fixation ensured a harmonic orientation of the phantoms across different acquisitions. The first holder contained four kiwis, while the second contained the tomatoes and onions. A pre-selected kiwi was rotated perpendicularly to its primary axis’s (labeled by kiwi_rot_) between its three repetitions to examine the possible influence of the phantom orientation on the computed RI [36,37,38,39].

### 2.2. 3D Printed Phantoms

Two types of distinct, hollow plastic objects were planned and constructed. First, a Quick Response (QR) code was modeled within a fillable container, including the “UNIDEB MRI Texture Analysis Phantom” text information. We used a web application for QR code production (https://www.qrcode-monkey.com/#text (accessed on 6 May 2021)). The QR code was printed into two 3D models with different sizes (large QR (5 × 5 × 4 cm^3^) and small QR (4 × 4 × 3 cm^3^), with respective heights of the QR code of 3 and 2 cm) using the Trimble SketchUp Pro 2020 (Trimble Inc., Sunnyvale, CA, USA). Both cubic QR code containers were constructed and saved in Standard Tessellation Language (STL) file format (Figure 2, first row) [33,34,40,41].

As the second type of printed phantom, we chose a 3D Hilbert cube with a volume of 5 × 5 × 5 cm^3^ [42]. The recursive process fills the entire space with one continuous line. We used an STL file of a pre-created Hilbert cube from the Thingiverse website to embed it to the mentioned volume (https://www.thingiverse.com/thing:1762713 (accessed on 10 June 2021)). From the downloaded STL files, the second recursion level was chosen due to its sufficient complexity and ease of redesign. We redesigned the original model containing several small vertical and horizontal support lines to achieve a more robust form to ensure successful printing and make an extra chessboard-like pattern in three dimensions. This model has an outside pattern with a plastic-air alternating rectangle and includes the so-called Hilbert square pipe (Figure 2, second row).

All three phantoms were 3D printed from Polylactic Acid (3DJake ecoPLA, White) filament on a Creality Ender 3, Fused Deposition Modeling (FDM) type 3D printer with a 0.4 mm nozzle diameter [27,30]. The print plan was achieved using Repetier-Host (Hot-World GmbH and Co. KG, Willich, Germany) software with the following settings: 100% infill; 0.2 mm layer height; no support; brim adhesion; 40 mm/s print speed; 210 °C hot end, and 60 °C print bed temperatures. The print time was around 4 h for each object [30,43]. Each 3D printed phantom was filled with 0.01 mM NiCL_2_ solution. Representative MR images of the QR and the Hilbert cube phantoms are presented in Figure 3.

### 2.3. MR Scanning

All phantoms were scanned at the MR imaging facility of the Clinical Center, University of Debrecen. The phantoms were scanned in two MR systems, a 3 T Philips Achieva and a 1.5 T Siemens Magnetom Essenza MRI scanner. The scans were performed with a 6-channel head coil in the 1.5 T MR and an 8-channel head or a 32-channel neurovascular coil in the 3 T MRI system. In both devices, isotropic 3D T2-weighted and 3D T1-weighted sequences were obtained using protocols employed in the clinical routine, with minor adjustments to the number of repetitions to ensure sufficient image quality in the smaller volumes. In the Philips MR system, the T1- and T2-weighted measurements were obtained using a 3D BrainVIEW protocol which provides high-resolution isotropic data. For the Siemens system, the SPACE (Sampling Perfection with Application optimized Contrasts using different flip angle Evolution) sequence was applied for T2 acquisitions, which provides isotropic 3D images. For T1-weighted Gradient-Echo (GRE) measurements, the sequence MPRAGE (Magnetization-Prepared Rapid GRE) was utilized [44,45,46].

The acquisition parameters for the acquisition protocols are listed in Table 1. Each imaging protocol was performed using two different isotropic voxel resolutions (1 × 1 × 1 mm^3^ and 2 × 2 × 2 mm^3^). The phantoms were scanned thrice for each setting in both MR systems, denoted as repeated scans or repetitions below. Specific to the Philips MR system, a new table position had to be chosen before each repetition. After completing imaging protocol, the acquisition software automatically moved the table from the gantry. However, the phantoms were always placed in the isocenter, the most homogeneous magnetic field region.

### 2.4. Image Visualization and Segmentation

A radiologist and a radiologist MR specialist with 30 and 10 years of MRI experience, respectively, analyzed the image data qualitatively in 3D Slicer software [47,48] to determine how close the imaged texture of each object was to the actual pattern and heterogeneity.

Image alignment was performed semiautomatically using the 3D Slicer open-source software (version 4.10.2 r28257) [49]. A cubic volume of interest (VOI) was placed separately on each 3D printed phantom using the CreateModels tool from the SlicerIGT and the Segmentations modules. VOI sizes were 50 × 50 × 50, 55 × 55 × 55, and 45 × 45 × 45 mm^3^ for the Hilbert cube, the large QR, and the small QR phantoms, respectively. The “Grow from seeds” algorithm was applied for biological phantoms to define VOIs that blend into the surface of fruits and vegetables as much as possible. The semiautomatically generated VOIs were manually corrected by excluding the border zone between fruit/vegetable and surrounding air and the most apical and basal slice of each fruit/vegetable using a brush-erase tool. Corresponding label maps from segmentations of all images were exported and saved to Neuroimaging Informatics Technology Initiative (NIfTI) file format for further processing and calculation [50,51].

### 2.5. Normalization and Discretization

In this study, we used a so-called µ ± 3σ normalization technique with µ being the mean and σ the standard deviation of the image [52,53,54,55,56]. Two different discretization techniques were considered; the *FBS* and the *FBN* techniques. The *FBS* method is defined as
$$I_{FBS}\left(i\right)=\left[\frac{I\left(i\right)}{B}\right]-1$$
where *I*(*i*) and *I_FBS_*(*i*) are the original and the transformed intensity level of the ith voxel [53,54,57,58]. The [] brackets stand for the ceil operation. We choose *B* = 0.15 for normalized images and *B* = 50 for non-normalized ones to have a similar number of bins in both cases [7,12,59].

The *FBN* method is calculated by
$$I_{FBN}\left(i\right)=\left\{\begin{array}{cc} 1 & I\left(i\right)=I_{min}\\ \left[D\cdot \frac{I\left(i\right)-I_{min}}{I_{max}-I_{min}}\right] & otherwise \end{array}\right\}$$
where *I_FBN_*(*i*) is the new intensity value of the *i*-th voxel intensity after the *FBN* discretization, *I_max_* is the maximum, *I_min_* is the minimum original voxel intensity of the particular lesion, and *D* is the number of bin parameters. We set *D* to 64.

### 2.6. Texture Calculation

RIs were extracted using the GLCM (Gray Level Co-Occurrence Matrix), GLSZM (Gray Level Size Zone Matrix), and GLRLM (Gray Level Run Length Matrix)-based algorithms implemented in MATLAB (version 2020) [7]. A total of 40 radiomic features were extracted from each VOI, divided into 18 GLCM, 11 GLSZM, and 11 GLRLM metrics. All 40 RIs (Appendix A) were calculated according to the IBSI guideline [53,59]. In addition, five basic histogram-based statistical parameters were also determined as the minimum and maximum value, mean, median, and VOI volume in voxel numbers. All 45 features were determined for the segmented volumes for each acquisition and discretization setup (Table 2).

## 3. Statistical Analysis

### 3.1. Coefficient of Variation

For each *RI* and each acquisition, we report the mean, standard deviation, and coefficient of variation (*CV*), defined as
$$CV=\frac{std\left(RI\right)}{mean\left(RI\right)}\cdot 100$$
where *std*(*RI*) and *mean*(*RI*) represent the standard deviation and mean for the three computed radiomics indices. We utilized the *CV* parameter as a measure of the repeatability of a group or object [9,18,60,61].

### 3.2. Relative Parameter Differences

The relative parameter difference (*RPD*) value was employed to measure the relative *RI* difference of a given phantom between two different measurement setups. The relative parameter difference was defined as:$$RPD=\left|\;\frac{mean(RI_{setup1})-mean(RI_{setup2})}{mean(RI_{setup1})}\right|\cdot 100$$

With $mean(RI_{setup1})$ and $mean(RI_{setup2})$ being the mean of the three *RI* at two different acquisition setups (for the definitions of acquisition setups, see Table 2).

Table 3 shows all the acquisition setup pairs and the related abbreviation used in the comparative analysis for each phantom type for T1 contrast. For the T2 contrast, the number of comparisons was the same, giving a total of 14 comparisons for both contrasts.

### 3.3. Interclass Correlation Coefficient

For the repeatability test, the interclass correlation coefficient (*ICC*) for two-way mixed effects [62] was calculated for each *RI* based on absolute agreement, single rater/measurement model. The *ICC* was determined by matching results (averages of the repeated measures) from two different measurement setups and using every phantom with the following formula:$$ICC=\frac{MS_{R}-MS_{E}}{MS_{R}+\left(k-1\right)MS_{E}+\frac{k}{n}\left(MS_{C}-MS_{E}\right)}$$
where $MS_{R}$ stands for mean square for rows, $MS_{E}$ is the mean square of error, $MS_{C}$ the mean square of columns, *n,* and *k* are the numbers of subjects and the number of raters/measurements. Based on the ICC-values the RI were considered to be either excellent (*ICC* > 0.9), good (0.75 < *ICC* ≤ 0.9), moderate (0.5 < *ICC* ≤ 0.75) and poor (*ICC* ≤ 0.5) repeatability [21,63,64,65,66]. Every calculation was performed using MATLAB (2020, The MathWorks Inc., Natick, MA, USA) and Microsoft Office Excel software.

### 3.4. QR Code Readability Test

The QR phantoms were evaluated using two analyses, the QR codes readability in the MR scans and the radiomics analysis of the texture. The readability of the QR code was tested for each of the coronal 2D images. An unsuccessful read-out of the QR code for any coronal slice was defined as a significant image distortion (>25% data loss) and thereby detrimental texture loss. The reading of the QR codes relied on two types of decoding methods; first, we wrote a Python program, while the second decoding was obtained using cell phone readings of the MR images. For the Python implementation, we utilized the Pyzbar module (https://pypi.org/project/pyzbar (accessed on 29 June 2021)) to read the coded textural information. The Pyzbar module ensures that multiple QR codes can be decoded simultaneously from an image with multiple QR codes. The original DICOM images must be converted to Portable Network Graphics (.png) format for this process. The resulting .png images will be mentioned as ‘original images’ in this manuscript. The original images were processed further and resized to 1024 × 1024-pixel resolution using the Lanczos interpolation method implemented in the Python Pillow library to ensure the read-out yield stability. The interpolated, 2D images will be referred to as ‘interpolated images’ below. The evaluation program reads every coronal image from a scan of QR code phantoms and tries to read out each coded information. The number of successful readings was expressed as a percentage of the total number of coronal images, and this parameter was defined as the reading ratio. For the second read-out method, we used commercial mobile phones to detect the QR codes from the same monitor screen (ViewSonic VP2030B monitor with 100% luminance and 1600 × 1200-pixel resolution) displaying the previously constructed PNG images of the QR phantoms [67,68,69,70]. The displayed size of each PNG image has been set to display the QR phantoms in their original, true size. Five phones were involved from three manufacturers: iPhone 12, iPhone SE 2020, Samsung Galaxy A51, Xiaomi Redmi 6A, and Xiaomi MI 9 lite, referred to respectively as Phone 1 through Phone 5 below. From a given scan, only the middle coronal image was processed by each phone.

## 4. Results

### 4.1. Visual Comparison

Figure 4A shows representative images of the large QR and the Hilbert phantoms at T1 weighted contrast and different acquisition resolutions (1 mm on the left and 2 mm on the right). We report a robust read-out of the Hilbert cube disregarding the image resolutions. In contrast, significant changes in the high-frequency patterns were reported for the QR cubes when the image resolution was changed (Figure 4C). After visual inspection of the high-resolution and low-resolution images, a radiologist and a radiographer MR specialist reported that the textures observed in the biological phantoms were comparable to the real heterogeneity. However, the finer structures (high-frequency features) were blurred (Figure 4B,D).

### 4.2. Coefficient of Variation

For repeatability purposes, CVs were determined using normalized and non-normalized data from each object with each MR acquisition setup (Appendix A). Improved repeatability (lower CV) measures were observed for all phantoms following the normalization of the data (Table 4 and Table 5).

Below, we present only results from the normalized data. Given the poor repeatability (CV > 10%), a subset of RIs were excluded from the subsequent analyses (Jmax, Energy, ClusterShade, HGRE, SRHGE, LRHGE, LZE, LZLGE, LZHGE).

The CVs for the remaining 31 RI were obtained using the different weighting (T1 and T2), and their repeatability coefficients are shown in Figure 5, Figure 6 and Figure 7 and Appendix A. Table 6 comprises the corresponding CV_average_ values for different property groups (Acquisition setup, Texture parameter, and Discretization method). In general, it may be observed that most texture features’ repeatability was better at 1.5 T field strengths than at 3 T (CV_average_ is 1.94 at 1.5 T and 2.11 at 3 T) (Figure 5, Figure 6 and Figure 7 and Table 4). In addition, no major differences were observed between the FBN and FBS discretization (CV_average_(FBN) = 2.04 and CV_average_(FBS) = 2.06).

Each object’s CV result from different points of view is presented in Table 6. The choice of discretization did not affect the repeatability measures (Table 6). In the texture parameters group, increased histogram-based CV values can be observed at the kiwis compared to the other objects’ same calculations. This phenomenon was not identified for the kiwi_rot_ (kiwi rotated through its primary axis between measurements). Columns of the acquisition setup group reveal that CV values are usually smaller at 1.5 T field strength and 1 mm resolution. Further, when excluding the kiwis, no difference in the repeatability was observed between the biological and 3D phantoms. Detailed CV data for all objects and acquisition setups are presented in Figure 5, Figure 6 and Figure 7 heatmaps.

Figure 7 shows the sensitivity of the radiomics when the scanned object rotates in the MR system. The CV calculated from the three orthogonal orientations (kiwi_rot_) is larger than the CVs obtained from three repetitions of a given kiwi in the conventional acquisition orientation.

### 4.3. Relative Difference

The relative parameter difference (RPD) values are shown in the Appendix A. The biggest relative differences are observed when comparing results obtained at different MR field strengths.

### 4.4. Interclass Correlation Coefficient

Figure 8 represents all ICC data of RI for the same comparison types as in the RPD analysis (in the Appendix A). The high ICC correlates well with the larger RPD values for each object and RI.

### 4.5. QR Code Readability Test

QR code readability outcomes show apparent dependency on the acquisition resolution with a standalone Python code and mobile phones. None of the applied read-out methods could detect the information from the 2 × 2 × 2 mm^3^ resolution scans, and only some of the 1 × 1 × 1 mm^3^ resolution images could be successfully read. Read-out ratios originating from the Python code-based process are presented in Figure 9. Ratios are calculated from the three repeated measurements as the average read-out ratio for each type of measurement.

Mobile phone read-out numbers are shown in Table 7. The related score distribution is similar to the result shown in Figure 9. Many of the T1 images of the QR cubes were readable by the phones.

Representative QR code images are shown in Figure 10 at T1 and T2 weighting and 3T field strength.

## 5. Discussion

This study aimed to test the feasibility of using printed 3D phantoms with structural information to analyze the robustness and repeatability of radiomics in MR imaging. To this end, this was performed using three different types of printed 3D phantom models and three types of biological phantoms (kiwis, tomatoes, and onions). The main finding of this study was that 3D printed phantoms provide similar robustness and repeatability metrics as biological phantoms and thus, may be favorable in identifying the optimal radiomics in MR imaging protocols. Of note, for high-resolution MR images (isotropic volume of 1 × 1 × 1 mm^3^), it is possible to retrieve the structural QR information, thus providing acceptable image quality for radiomic analyses. To the best of our knowledge, this finding has not been previously demonstrated in studies evaluating radiomics.

Radiological assessments in tumor staging are primarily focused on the number and the size of lesions in radiological settings when using MR; nevertheless, there is also a growing interest in measuring and analyzing radiomic characteristics, which may provide more insight into the lesion composition [71,72,73,74,75,76,77,78,79,80]. Furthermore, it is not fully understood in all its details how the different MR systems and acquisition protocols affect the robustness and reliability of radiomics parameters [64,81,82,83,84]. In general terms, radiomic assessments are challenged by the non-quantitative nature of MR imaging. The intensities in the MR images may be affected by shimming protocols and positioning of the scanned objects, among other factors, which may pose challenges in the test-retest assessment of the radiomic information [77,85]. In addition to the standard imaging parameters such as field of view, spatial resolution, and reconstruction algorithm, other factors such as magnetic field strength, repetition number, echo time, number of excitations (NEX or NSA), or the signal-to-noise ratio itself have a high impact on calculations [86]. Nevertheless, there is also a growing interest in measuring and analyzing radiomic characteristics [71,72,73,74,75,76,77,78,79,80]. One way to introduce radiomics into MR imaging protocols and to identify the relevant metrics for MR images may be through phantom studies. Previous studies have focused on biological phantoms, showing that fruits and vegetables are suitable as phantoms because they have sufficient water and organically fine texture [15,16,36,38,87]. In addition, Werz et al. showed T1 and T2 relaxation times for fruits and vegetables are similar to human tissues [36]. They found that the measured image quality of biological objects has good reproducibility, making them useful for test measurements, sequence developments, and optimization. Despite the positive findings as potential phantoms, the fruit and vegetables have limited applicability in the multi-center studies of radiomics because of their relatively rapid biological decay. Therefore, this study sought to identify if 3D printed phantoms may act as replacements. Figure 5 and Figure 6 and Table 4 and Table 6 showed that the 3D-printed phantoms provided repeatability metrics similar to those obtained from the biological phantoms. Further, it was found that the discretization process did not affect the repeatability metrics.

In radiomics, it is important to identify the most relevant metrics for tumor characterization. The literature and IBSI guidelines have identified several hundred radiomic metrics that may be extracted from segmented VOIs in medical images [1,3,53,54,57,88,89,90,91]. While all metrics may contribute to the lesion assessment, many may be relevant only for a subset of organs and tumor phenotypes, while others may have imaging modality-specific performances. Therefore, a pre-selection of the metrics used for the initial radiomic assessments may improve both the reliability of the identified metrics and the overall processing speed of the applied machine-learning models [21,80,92,93]. Our proposed 3D printed phantoms may suit such purposes.

Several studies of biological phantoms have proven that the segmentation procedure may affect texture analysis; hence we chose two robust volume delineation methods: the fixed cube-shaped VOI definition and the “grow from seeds” 3D Slicer tool [7,47]. Figure 4A demonstrates that the visible robust texture of the Hilbert cube is not affected by the spatial resolution when using 1 mm and 2 mm isotropic volumes. However, the image structure of the finer patterned QR cubes deteriorates at 2 mm spatial resolution for both the large QR and small QR cubes (Figure 4A,C). Similar findings were reported for the biological phantoms (Figure 4B,D), where high-frequency objects such as the seeds were smeared. These results emphasize that the performance level of RI is highly dependent on the spatial resolution of the input images.

The radiomic analyses’ repeatability is important in translating their use into the clinical routine. In this study, we investigated the repeatability measures at different steps in the analyses, the normalization process, and the RI, in addition to how they were affected by object size and imaging parameters. We report improved repeatability for both biological and printed phantoms when normalization is applied to the data (Appendix A, Table 4), findings that are in concordance with previous studies [7,52,56,94,95,96,97]. Similarly, when normalization was applied to the data, improvements were observed for the two discretization methods (FBS and FBN). This finding is concordant with previous findings for the FBS normalization [56,71,94,98]. Similarly, the radiomics parameter groups (Table 5) had improved repeatability when normalization was applied to the data, stressing the importance of using normalization of MR images before radiomic analyses. Despite normalization, we observed that a subset of RIs was performing poorly (CV > 10%) for all MR imaging protocols and objects. Because of their poor performance, Jmax, Energy, ClusterShade, HGRE, SRHGE, LRHGE, LZE, LZLGE, and LZHGE were omitted from further analyses in this study, and only normalized data were used in all following evaluations. The resulting RIs were observed to perform better at lower MR field strengths, in agreement with previous reports (Figure 5, Figure 6 and Figure 7, Table 6, and Appendix A), findings likely caused by worsened field homogeneity at 3 T [10].

In terms of parameter performance, we observed that the relative difference (RPD) might change by more than 20% for the same objects, depending on the MR systems’ field strengths (Table 3, Appendix A, and Figure 6). Further, a bias was observed for the RPD when altering the imaging resolution, regardless of the image weighting (T1 or T2) and phantom type. Based on RPD comparisons, it can be concluded that, from the applicability point of view, 3D printed phantoms are as good as biological ones. It should also be noted that artificial phantoms eliminate some problems related to biological properties, such as putrefaction.

In terms of repeatability, high test-retest variance (high CV, poor repeatability) was observed for the mean and median values obtained in the objects (Table 6). The poor repeatability measures are likely introduced by the shimming the B0 field before each scan (introducing small alterations in signal intensities) and the non-quantitative intensities of the MR images [65,93,99,100]. The ICC was calculated by averaging the RI from the three repetitions of a given setup and matching these results between two different measurement setups (Table 3) involving all phantoms. Figure 8 shows the calculated ICCs of RI for the same comparison types as in the RPD analysis (Appendix A). The expected negative correlation of the ICC and RPD values is clearly visible within each parameter group. In general, a larger ICC corresponds to a smaller RPD value. Results of the manufactured QR cubes show how to degrade the stored deep-level information by MR imaging. Applying different coils, field strength, or even acquisition resolution influences the level of information loss [10,15,71,72,101]. The metric of the degradation of the embedded information could be used as a new, useful parameter besides the existing radiomics characteristics.

In this study, the quality of the texture information analysis was also evaluated by the read-out yield of QR codes. It was hypothesized that a successful read-out of the QR code meant that the MR image series preserved most of the physical properties of the original texture. Using Python read-out of the QR codes from the converted .png files, we report read-out successes < 5% for the small QR when using the T2 sequence at 3 T, while no read-outs were possible for the original (non-interpolated images) at 1.5 T (Figure 9). However, at 3 T, the original and interpolated large QR phantom may be decrypted when using the T1-weighted images. These findings were reproduced by reading the QR code with smartphones (Table 7). In general, the read-out success rates were of varying success, with read-out percentages ranging from 19% to 39%. The smartphone read-out success rates highlight that the successful decoding of stored information depends on the acquisition settings and the reader algorithm. This phenomenon is analog to the clinical situation where lesion detection may depend on the physician’s practice.

The 3D printed phantoms developed in this study have many advantages and could be helpful in the quality assurance of radiomic studies involving MR datasets. Printed phantoms using the same materials and print settings at the local sites are an easy and affordable method to compare the radiomics performance of different MR scanners. For the analysis of radiomic characteristics, the flexibility of 3D printing could be a favorable method. The newly proposed 3D printed phantoms and the results with the biological phantoms of this study may benefit the radiomics community, which seeks to standardize both imaging protocols and radiomic analysis strategies [21,43].

Our study has some limitations. First, the results from phantom studies cannot always be transferred directly into clinical studies. In clinical studies, the patients are known to move during the acquisition protocols, these patterns cannot be reproduced by simple 3D-printed phantoms examined in this study [92,102]. Therefore, the results obtained here simulate a best-case scenario where no patient motion is observed. Unlike analyses solely based on 3D printed and biological phantoms, results based on actual human tissue can better indicate the usefulness of radiomics. This is because whatever phantom is proposed, they are always a simplification of a real human tissue environment [35,38,103,104]. Second, we examined only the 3D radiomic characteristics of certain texture classes. However, we wanted to include only the most frequently used ones in this work [13,35]. In addition, the number of MR devices and sequences included in the study was also limited.

## 6. Conclusions

We report good agreements between observations obtained from biological and 3D-printed phantoms. Three-dimensional-printed QR codes provide a unique opportunity to analyze the reliability and challenges of radiomics in MR imaging protocols. This study found that the large QR phantom provides better insight into identifying radiomic features of interest than the usual Hilbert phantom. Further, the QR codes permit analyses of texture distortion through external validation of the readability of the QR codes using smartphone read-out success rates.

## Figures and Tables

**Figure 1 diagnostics-12-02196-f001:**
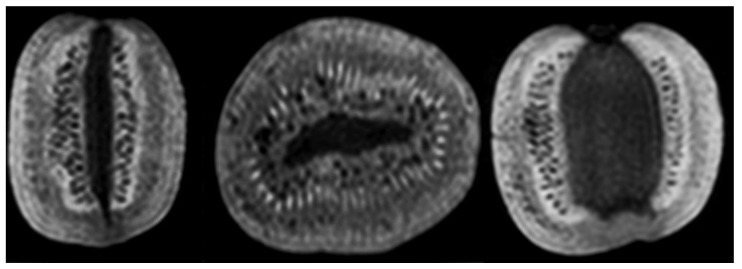
Representative orthogonal 3T MR images of a kiwi: 3D T1 weighted sagittal, coronal, and axial high resolution (1 × 1 × 1 mm^3^).

**Figure 2 diagnostics-12-02196-f002:**
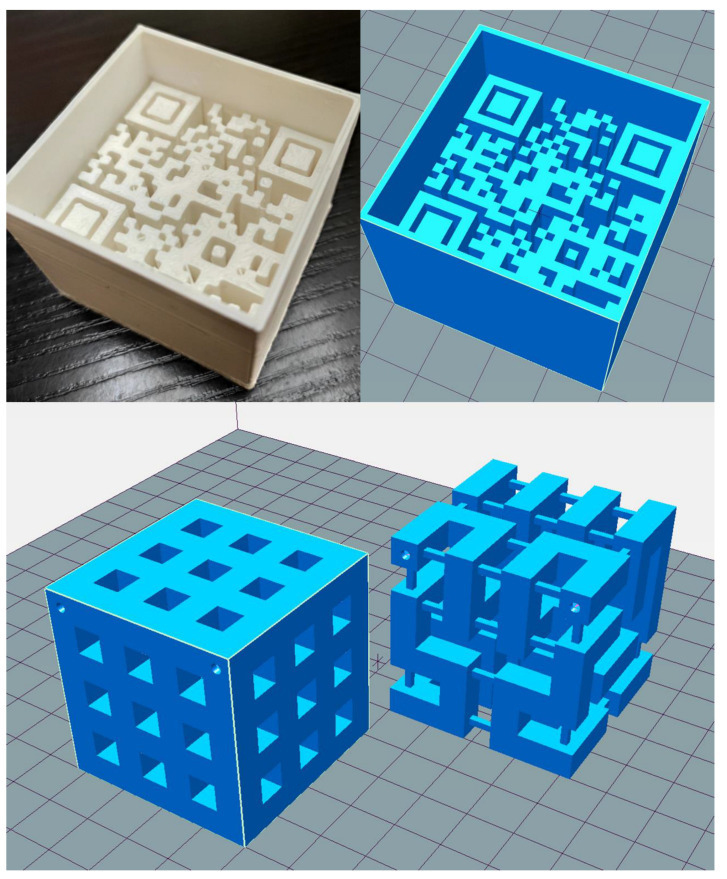
Representative photo and model images of the 3D printed phantoms: the top row shows a photo and the design image of the QR code, while the bottom row shows the design images of the modified Hilbert cube after and before modeling from left to right.

**Figure 3 diagnostics-12-02196-f003:**
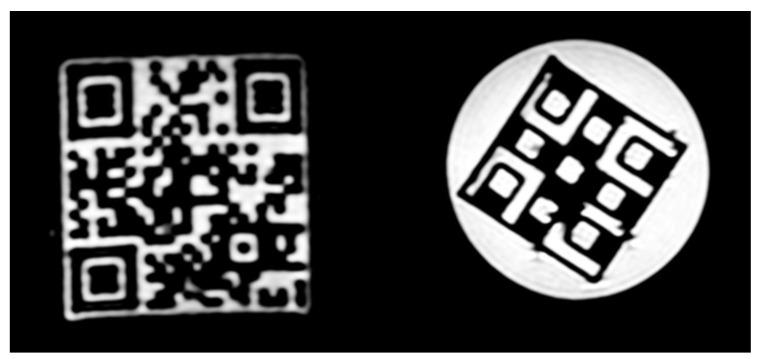
Representative coronal MR images of 3D printed phantoms. The left is the large QR cube, and the right is the Hilbert cube. The interested readers can read the presented QR code MR image using their smartphone’s QR code reader.

**Figure 4 diagnostics-12-02196-f004:**
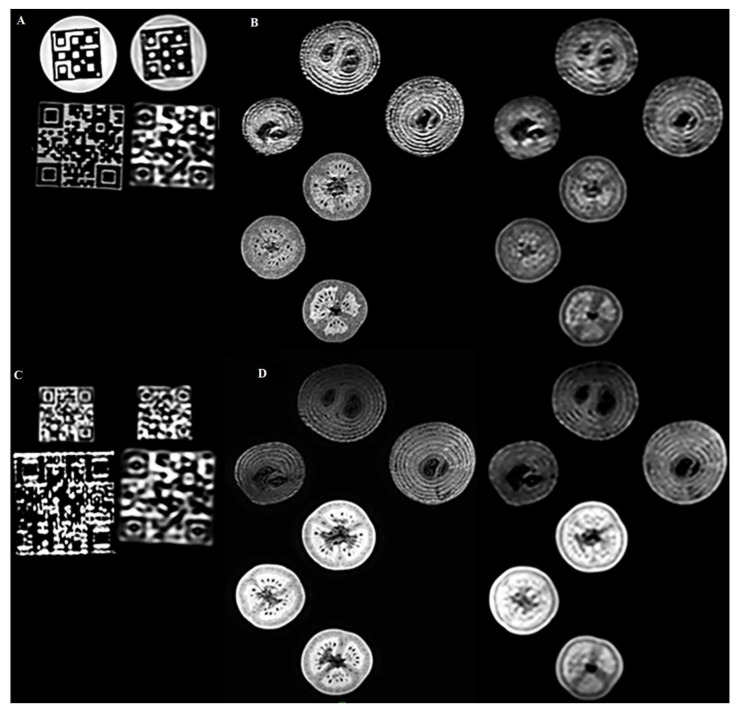
Representative 3D T1- and T2-weighted 3T MR coronal images of the 3D printed and biological phantoms. The pairwise left and right images had isotropic resolutions of 1 × 1 × 1 mm^3^ and 2 × 2 × 2 mm^3^, respectively. (**A**): 3D T1-weighted 3T MR coronal images of the Hilbert cube and the large QR phantoms. (**B**): 3D T1-weighted images of 3 tomatoes and three onions. (**C**): 3D T2-weighted 3T MR coronal images of small and large QR code phantoms. (**D**): 3D T2-weighted images of the same three tomatoes and three onions.

**Figure 5 diagnostics-12-02196-f005:**
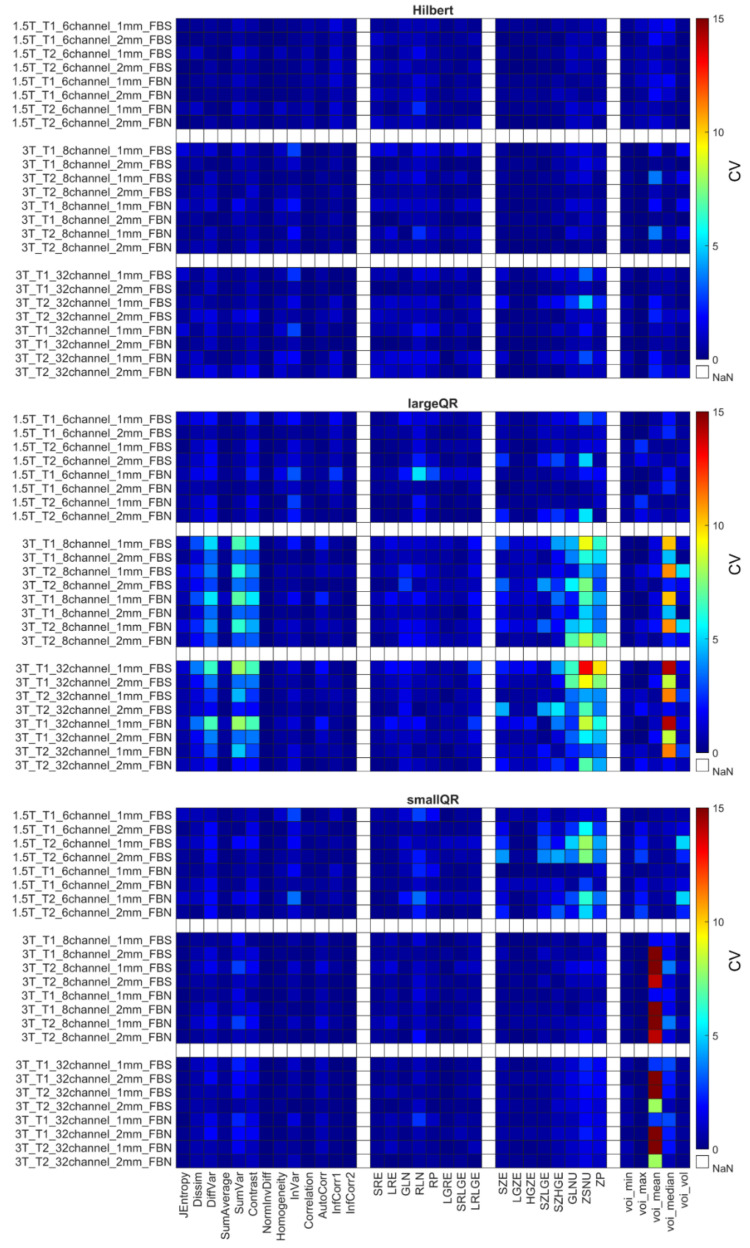
CV of radiomics parameters for Hilbert, large QR, and small QR code cubes. Four radiomics parameter groups (GLCM, GLRLM, GLRLM, and histogram-based) are shown on the horizontal axis. The CV is expressed in %.

**Figure 6 diagnostics-12-02196-f006:**
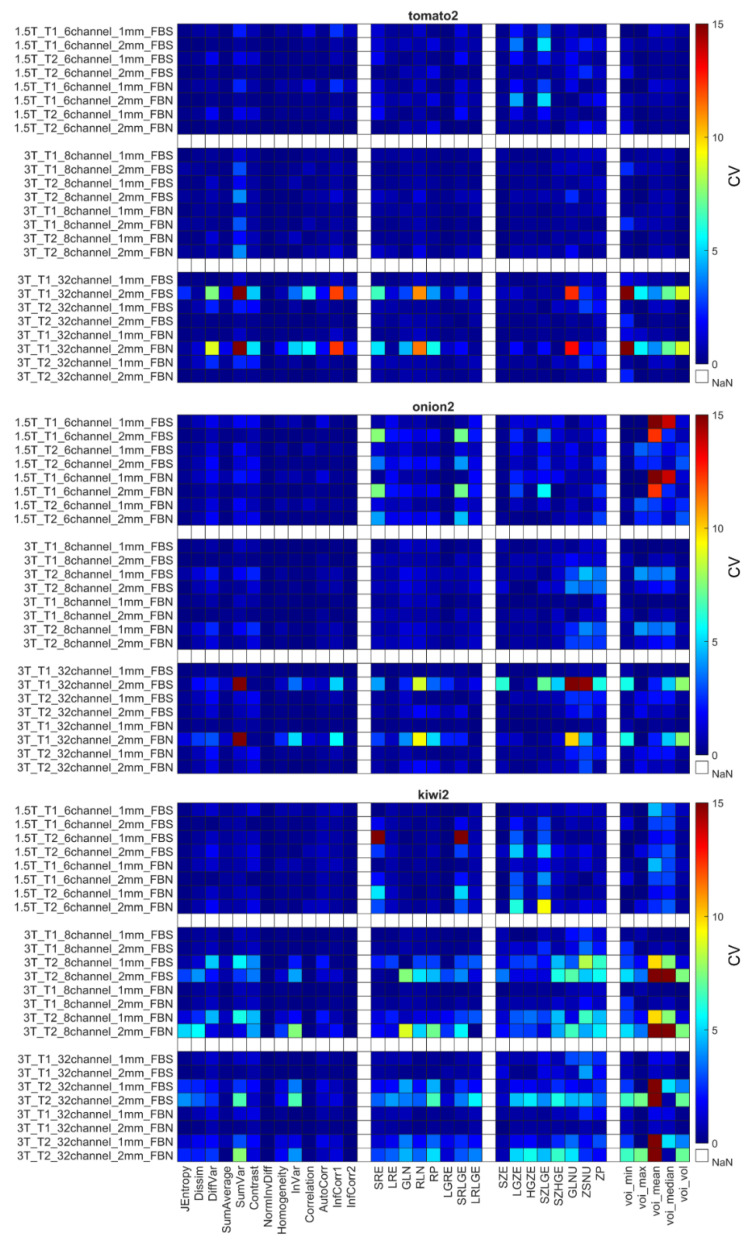
CV of radiomics data obtained for the second tomato, onion, and kiwi. Corresponding figures for all phantoms can be found in the Appendix A. The CV is expressed in %.

**Figure 7 diagnostics-12-02196-f007:**
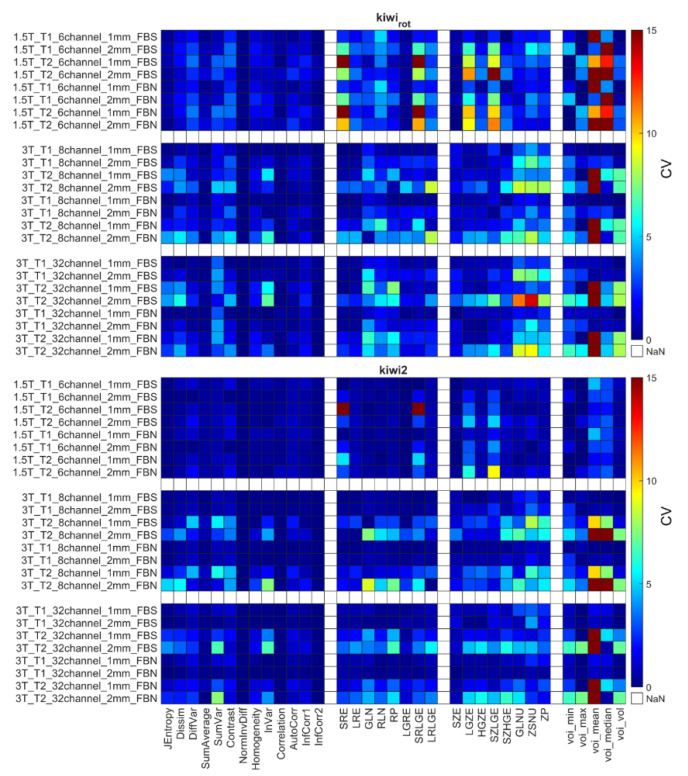
CV data for kiwi_rot_ and kiwi2 objects. The CV is expressed in %.

**Figure 8 diagnostics-12-02196-f008:**
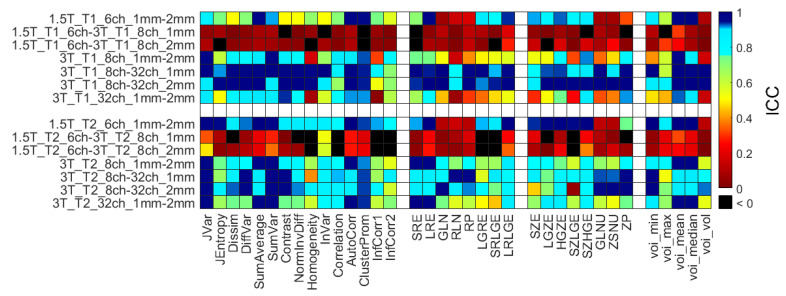
Heatmap of the ICCs calculated for the 14 different comparisons and computed RI. Black squares correspond to ICCs with a value below zero, indicating high error rates between the measurements. ICC = interclass correlation coefficient.

**Figure 9 diagnostics-12-02196-f009:**
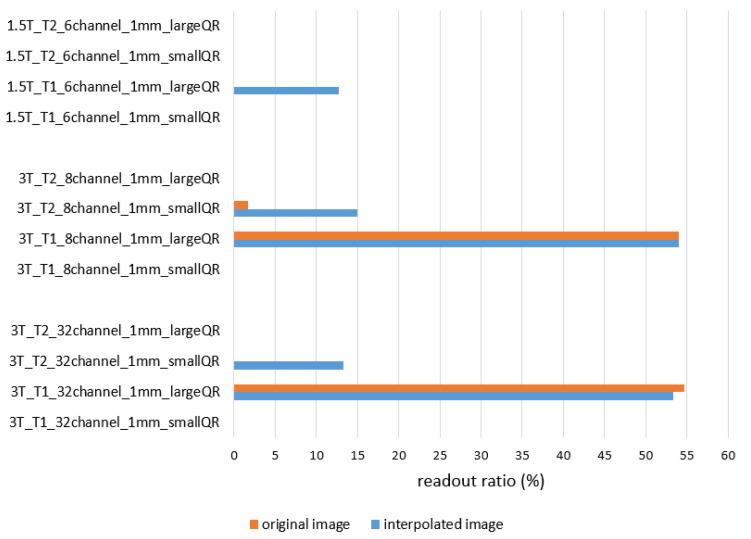
Averaged read-out ratios of the three repeated measurements in the case of the Python program processing. All original and interpolated coronal images of small QR and large QR phantoms were read by the developed Phyton code. Data from the 2 × 2 × 2 mm^3^ resolution are not presented due to the lack of successful decoding.

**Figure 10 diagnostics-12-02196-f010:**
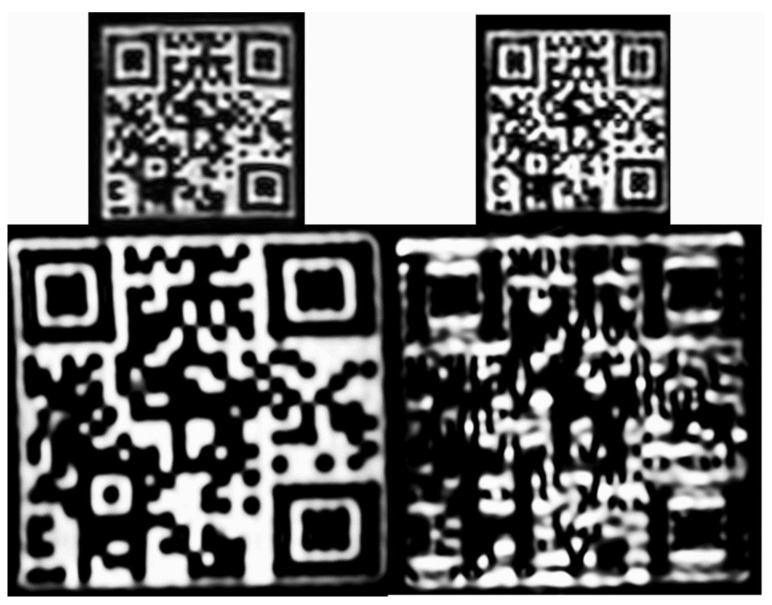
Representative coronal images from the small and the large QR phantom scans at 3 Tesla field strength and 1 × 1 × 1 mm^3^ resolution. T1 and T2 contrast images are in the left and right columns, respectively. The encoded information can only be read out with a smartphone from the T1 image of the large QR phantom.

**Table 1 diagnostics-12-02196-t001:** Parameters of the sequences used in this work. TR = repetition time, TE = echo time, NSA = number of signals averaged.

MR	Sequences	TR (ms)	TE (ms)	NSA	Voxel Size (iso mm)
3 Tesla	3D T2 BrainVIEW	2500	233	3	1, 2
	3D T1 MPRAGE	600	28.3	2	1, 2
1.5 Tesla	3D T2 SPACE	1200	97	2	1, 2
	3D T1 MPRAGE	1040	4	2	1, 2

**Table 2 diagnostics-12-02196-t002:** Abbreviations for the different imaging setups. The second column shows the setup parameters, listing the filed strength, the weighting (T1 or T2), the applied coil type, the acquisition voxel size, and the discretization method. A total of 24 settings were considered for each RI calculation.

#	Abbreviation	Field Strength	Weighting	Number of Channels	Voxel Size (mm^3^)	Discretization
1	1.5T_T1_6ch_1mm_FBS	1.5 T	T1	6	1 × 1 × 1	FBS
2	1.5T_T1_6ch_1mm_FBN	1.5 T	T1	6	1 × 1 × 1	FBN
3	1.5T_T1_6ch_2mm_FBS	1.5 T	T1	6	2 × 2 × 2	FBS
4	1.5T_T1_6ch_2mm_FBN	1.5 T	T1	6	2 × 2 × 2	FBN
5	3T_T1_8ch_1mm_FBS	3 T	T1	8	1 × 1 × 1	FBS
6	3T_T1_8ch_1mm_FBN	3 T	T1	8	1 × 1 × 1	FBN
7	3T_T1_8ch_2mm_FBS	3 T	T1	8	2 × 2 × 2	FBS
8	3T_T1_8ch_2mm_FBN	3 T	T1	8	2 × 2 × 2	FBN
9	3T_T1_32ch_1mm_FBS	3 T	T1	32	1 × 1 × 1	FBS
10	3T_T1_32ch_1mm_FBN	3 T	T1	32	1 × 1 × 1	FBN
11	3T_T1_32ch_2m1m_FBS	3 T	T1	32	2 × 2 × 2	FBS
12	3T_T1_32ch_2mm_FBN	3 T	T1	32	2 × 2 × 2	FBN
13	1.5T_T2_6ch_1mm_FBS	1.5 T	T2	6	1 × 1 × 1	FBS
14	1.5T_T2_6ch_1mm_FBN	1.5 T	T2	6	1 × 1 × 1	FBN
15	1.5T_T2_6ch_2mm_FBS	1.5 T	T2	6	2 × 2 × 2	FBS
16	1.5T_T2_6ch_2mm_FBN	1.5 T	T2	6	2 × 2 × 2	FBN
17	3T_T2_8ch_1mm_FBS	3 T	T2	8	1 × 1 × 1	FBS
18	3T_T2_8ch_1mm_FBN	3 T	T2	8	1 × 1 × 1	FBN
19	3T_T2_8ch_2mm_FBS	3 T	T2	8	2 × 2 × 2	FBS
20	3T_T2_8ch_2mm_FBN	3 T	T2	8	2 × 2 × 2	FBN
21	3T_T2_32ch_1mm_FBS	3 T	T2	32	1 × 1 × 1	FBS
22	3T_T2_32ch_1mm_FBN	3 T	T2	32	1 × 1 × 1	FBN
23	3T_T2_32ch_2mm_FBS	3 T	T2	32	2 × 2 × 2	FBS
24	3T_T2_32ch_2mm_FBN	3 T	T2	32	2 × 2 × 2	FBN

**Table 3 diagnostics-12-02196-t003:** List of abbreviations for calculating the *RPD* at T1 and T2 contrast. The first column contains the abbreviations, and the second and third columns show the serial number of the acquisition setup defined in Table 2.

Abbreviation	Compared Setups	
	Setup 1 (Row # of Table 2)	Setup 2 (Row # of Table 2)
1.5T_T1_6ch_1mm-2mm	1	3
1.5T_T1_6ch-3T_T1_8ch_1mm	1	5
1.5T_T1_6ch-3T_T1_8ch_2mm	3	7
3T_T1_8ch_1mm-2mm	5	7
3T_T1_8ch-32ch_1mm	5	9
3T_T1_8ch-32ch_2mm	7	11
3T_T1_32ch_1mm-2mm	9	11
1.5T_T2_6ch_1mm-2mm	13	15
1.5T_T2_6ch-3T_T2_8ch_1mm	13	17
1.5T_T2_6ch-3T_T2_8ch_2mm	15	19
3T_T2_8ch_1mm-2mm	17	19
3T_T2_8ch-32ch_1mm	17	21
3T_T2_8ch-32ch_2mm	19	23
3T_T2_32ch_1mm-2mm	21	23

**Table 4 diagnostics-12-02196-t004:** Repeatability is expressed via CV_average_, in the case of normalized and non-normalized images. At normalized images, lower CV values can be observed in most cases.

Object	without Normalization	with Normalization
Hilbert	1.55	1.22
smallQR	2.75	2.08
largeQR	1.90	1.86
tomato1	2.91	1.59
tomato2	2.75	1.21
tomato3	2.28	1.39
onion1	2.90	1.03
onion2	2.73	1.05
onion3	2.77	1.15
kiwi_rot_	4.82	4.40
kiwi1	2.98	3.87
kiwi2	3.60	0.46
kiwi3	1.73	2.71

**Table 5 diagnostics-12-02196-t005:** CV_average_ computed for each radiomics parameter group for all objects.

Radiomics Parameter Group	without Normalization	with Normalization
GLCM	3.08	2.72
GLRLM	2.90	2.04
GLSZM	3.87	2.26
Histogram based	3.09	6.11

**Table 6 diagnostics-12-02196-t006:** Average CV values for each object and different settings or properties. The last row shows the CV_average_ for the case when all objects’ data are included.

	Acquisition Setup						Texture Parameter				Discretization Method	
Object	1.5 T	3 T	1 mm	2 mm	T1	T2	GLCM	GLRLM	GLSZM	Histogram Base	FBS	FBN
Hilbert	0.60	0.75	0.85	0.55	0.71	0.69	0.55	0.64	1.09	0.64	0.67	0.73
largeQR	0.89	2.08	1.98	1.38	1.86	1.50	1.57	0.79	2.69	1.82	1.71	1.65
smallQR	1.08	2.04	1.21	2.23	2.17	1.27	0.66	0.55	1.30	7.46	1.76	1.68
onion1	0.73	2.04	1.48	1.73	1.60	1.61	1.49	1.05	1.51	2.98	1.61	1.60
onion2	1.50	1.38	1.03	1.80	1.80	1.03	1.26	1.17	1.46	2.22	1.46	1.38
onion3	1.23	2.03	1.43	2.09	1.78	1.75	1.37	1.26	1.60	4.00	1.74	1.78
tomato1	0.96	1.53	0.96	1.73	1.86	0.83	1.64	1.06	1.29	0.98	1.33	1.36
tomato2	0.72	1.61	0.65	1.98	1.99	0.65	1.75	0.78	1.11	1.23	1.32	1.32
tomato3	1.09	1.46	0.91	1.77	1.97	0.70	1.66	0.81	1.07	1.65	1.34	1.33
kiwi1	3.52	3.66	2.86	4.37	1.83	5.40	1.63	2.81	3.78	10.59	3.67	3.55
kiwi2	1.06	4.17	2.70	3.57	0.66	5.60	1.19	1.51	2.23	12.97	3.16	3.10
kiwi3	10.72	2.38	1.43	8.88	6.99	3.33	1.10	1.42	2.06	28.27	5.15	5.16
kiwi_rot_	1.15	2.28	1.43	2.38	0.69	3.12	1.23	1.31	2.04	4.64	1.91	1.90
All included	1.94	2.11	1.46	2.65	1.99	2.11	1.32	1.17	1.79	6.11	2.06	2.04

**Table 7 diagnostics-12-02196-t007:** Read-out results by different smartphones. The middle coronal images of the three consecutive measurements were used to show the ratio of the successful and total decrypting of the QR code information. Data from the 2 × 2 × 2 mm^3^ resolution are not presented due to the lack of successful decoding.

	# of Successful Read-Outs				
Read-Out Ratio	Phone 1	Phone 2	Phone 3	Phone 4	Phone 5
3T_T1_32channel_1mm_smallQR	2/3	-	-	-	-
3T_T1_32channel_1mm_largeQR	3/3	3/3	2/3	3/3	3/3
3T_T2_32channel_1mm_smallQR	-	-	-	-	-
3T_T2_32channel_1mm_largeQR	-	-	-	-	-
3T_T1_8channel_1mm_smallQR	2/3	-	-	-	-
3T_T1_8channel_1mm_largeQR	3/3	3/3	2/3	3/3	3/3
3T_T2_8channel_1mm_smallQR	-	-	-	-	-
3T_T2_8channel_1mm_largeQR	-	-	-	-	-
1.5T_T1_6channel_1mm_smallQR	-	-	-	-	-
1.5T_T1_6channel_1mm_largeQR	3/3	3/3	3/3	3/3	3/3
1.5T_T2_6channel_1mm_smallQR	-	-	-	-	-
1.5T_T2_6channel_1mm_largeQR	1/3	-	-	2/3	-

## Data Availability

The data presented in this study are available in the article and the corresponding Appendix A.

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
