# Peer review of "Phantom Study on the Robustness of MR Radiomics Features: Comparing the Applicability of 3D Printed and Biological Phantoms"

_diagnostics, 2022, doi:10.3390/diagnostics12092196_

Round 1
Reviewer 1 Report
3D printed phantoms in MRI is proposed in this paper. Hilbert cube, furthermore, and QR code cube were studied. The results show that the printed 3D Hilbert and QR code cubes have good agreement with the results of the biological phantoms. So, this paper should be accepted for publication.
Author Response
Dear Reviewer,
We sincerely thank the reviewer for his/her opinion, comments, and suggestion for acceptance.
Reviewer 2 Report
This is an interesting study involving the use of 3D printing to develop magnetic resonance imaging phantoms for radiomics studies. However, a number of issues are noted. The details are as follows. Please address all these comments accordingly.
1. The abstract has 434 words. According to the Instructions for Authors of the Diagnostics journal (https://www.mdpi.com/journal/diagnostics/instructions), “the abstract should be a total of about 200 words maximum.” Please shorten the abstract to make it compliant with the journal requirement.
2. Lines 121-131: Please cite references for the points there.
3. Lines 136-145: Please cite references for the points there.
4. Lines 194-195: Please fix the grammatical error.
5. Lines 203-212: Please cite references for the points there.
6. Lines 245-261: Please cite references for the points there.
7. Lines 274-276: Please cite references for the points there.
8. Lines 279-304: Please cite references for the points there.
9. Results, Visual comparison: It seems the corresponding method for the results there is not provided in the Methods section. Please add these method details into the Methods section and cite appropriate references to support it.
10. There are 19 illustrations in this manuscript which appears too many. You should move some of them to the Supplementary Materials section. I suggest the manuscript should only have a maximum of 10 illustrations.
11. Lines 434-449: It is stated that "3D printed, cube-shaped, heterogenous, and information-embedded phantoms did not appear in radiomics literature". Can you please provide substantial evidence for this point? Also, why no previous study explored this? Please provide further explanations with relevant reference support. Also, as suggested by you that no previous study has explored this, what is your basis to try this approach? Please further discuss this with literature support.
12. Lines 506-572: Please cite references for the points there. Also, please compare your study findings with those from other similar studies.
13. Discussion: The whole section only has one paragraph. Please divide the contents into multiple paragraphs for improving readability.
14. Lines 564-572: As per the content there, it seems your study value is limited. Can you please further justify your study limitations?
Author Response
Dear Reviewer,
We would like to thank the reviewer for the detailed evaluation and comments. We are grateful for the thorough review of our research. The reviewer's comments and questions provided an opportunity to improve the manuscript. Thank you for pointing out the typos and inaccurate English usage. The manuscript was revised in the suggested ways. As a result, please read our point-by-point response below. The reviewers’ comments are in bold and italic.
- The abstract has 434 words. According to the Instructions for Authors of the Diagnostics journal (https://www.mdpi.com/journal/diagnostics/instructions), “the abstract should be a total of about 200 words maximum.” Please shorten the abstract to make it compliant with the journal requirement
Thank you for pointing out that our abstract is over the word limit. We managed to shorten it to 216 words. However, the large number of methods and results does not allow for less. Please note that several abstracts of manuscripts published in the journal have more than 200 words.
2-8. Lines 121-131; 136-145; 203-212; 245-261; 274-276; 279-304: Please cite references for the points there.
Thank you for your comments, we have accordingly added the necessary references to the manuscript.
- Results, Visual comparison: It seems the corresponding method for the results there is not provided in the Methods section. Please add these method details into the Methods section and cite appropriate references to support it.
Thank you for your useful observation, we have accordingly modified a chapter entitled "Image visualization and segmentation". Additional references have been provided.
- There are 19 illustrations in this manuscript which appears too many. You should move some of them to the Supplementary Materials section. I suggest the manuscript should only have a maximum of 10 illustrations.
Thank you very much for your comment, 3 pictures from the manuscript have been moved to the supplementary material, leaving 10 pictures and 7 tables.
- Lines 434-449: It is stated that "3D printed, cube-shaped, heterogenous, and information-embedded phantoms did not appear in radiomics literature". Can you please provide substantial evidence for this point? Also, why no previous study explored this? Please provide further explanations with relevant reference support. Also, as suggested by you that no previous study has explored this, what is your basis to try this approach? Please further discuss this with literature support.
We thank the reviewer for the questions. Our statements about uniqueness indeed need to be proved.
To search for the following keywords “3D printing”, “radiomics”, “phantom”, and “MRI” in together, Google Scholar provide nearly 800 hits. Applying the same search condition for PubMed, ScienceDirect, and ResearchGate we got 3, 13, and 100 hits, respectively. These obviously overlapped. All our searches were performed in June of 2022.
Most of these studies only mentioned that 3D printing is a possible way to produce radiomics phantoms (10.1016/j.ejmp.2020.02.003). Others have 3D printed phantoms segmented from real patient data (10.1097/RLI.0000000000000530, 10.1016/j.ejmp.2022.04.007, 10.1016/j.urology.2017.08.056) or one with artificially created shapes (10.1002/mp.14173 ).
The aspiration is great to reach a phantom construction that is exact with the real biological data (https://doi.org/10.1155/2019/1071453) or to unify radiomic parameters based on the comparison of the results of the biological organs and their segmented and 3D-printed models.
Radiomics, especially in the field of MRI, is far from simple. Even small contrast differences could affect numerous radiomics parameters, and a uniform phantom solution is not accessible in the field (doi.org/10.3390/jpm11090842 ).
Creating structural or material heterogeneity with a 3D printer are two different approaches. In our study, structural heterogeneity was implemented by three 3D printed phantoms that have mathematically predefined heterogeneous shapes and textures.
However, our phantoms also have embedded text information that surpasses the simply modeled literature-based heterogeneous structures.
Our basic idea was not just to create heterogeneous 3D printed phantoms and acquire the radiomics features but also to study how the information stored in a real-world object is affected by the acquisition and the post-processes. Moreover, another idea was to compare our 3D printed objects with inherently different biological phantoms to prove that 3D printed phantoms can also be utilized.
We added new and structurally independent information to our phantom construction, which is a new parameter besides the non-unified existing ones. In our opinion, that is why no previous study did not explore this approach.
- Lines 506-572: Please cite references for the points there. Also, please compare your study findings with those from other similar studies
Thank you for your comment, we have accordingly added the necessary references and compared our findings with similar ones.
- Discussion: The whole section only has one paragraph. Please divide the contents into multiple paragraphs for improving readability
We divided the discussion section into several sections as it was suggested.
- 14. Lines 564-572: As per the content there, it seems your study value is limited. Can you please further justify your study limitations?
Additional limitations of our study have been inserted in the discussion section.

Reviewer 3 Report
The authors aimed to developed fillable Quick Response (QR) code- and Hilbert curve-based 3D printed models to introduce new MRI radiomic phantoms. Furthermore, they chose biological phantoms that have previously been found useful in the literature (kiwis, onions, and tomatoes), and measured them on the same systems to identify which 3D phantom type is the most suitable for testing the reliability of MRI radiomics features.
The study covers some issues that have been overlooked in other similar topics. The structure of the manuscript appears adequate and well divided in the sections. Moreover, the study is easy to follow, but some issues should be improved. Some of the comments that would improve the overall quality of the study are:
a. Authors must pay attention to the technical terms acronyms they used in the text.
b. English language needs to be revised.
c. Conclusion Section: This paragraph required a general revision to eliminate redundant sentences and to add some "take-home message".
Author Response
Dear Reviewer,
We respectfully thank the reviewer for the detailed evaluation and comments. The manuscript was revised in the suggested ways. As a result, please read our point-by-point response below. The reviewers’ comments are in bold and italic.
- Authors must pay attention to the technical terms acronyms they used in the text.
Thank you for your comment, we tried to standardize all technical terms and acronyms in the manuscript.
- English language needs to be revised.
In some cases, inappropriate English words and expressions were used in the manuscript. We improved the legibility of the manuscript.
- Conclusion Section: This paragraph required a general revision to eliminate redundant sentences and to add some "take-home message".
Thank you for your suggestion. We have reformatted and shortened the conclusion section to make the essence of our study more understandable. In addition, a "take-home message" was formulated in this section.

Round 2
Reviewer 2 Report
Thank you for the revision. However, I feel that some of my previous comments have not been adequately addressed yet. My detailed comments are as follows.
1. Abstract: please provide key figures to support your result description.
2. This manuscript still has 17 illustrations (10 figures and 7 tables) which appear too many.
3. Content organisation and flow of idea need improvement. For example, the second sentence of the Introduction section should be moved to the beginning. Please review the content organisation and flow of idea for the whole manuscript to improve readability. Some abbreviations, e.g. RI in line 84, etc. are not defined before using. Please address this issue.
4. The basis of using QR code pattern for printing the phantoms is still not clearly presented in the Introduction. Suggest providing further discussion about this in the Introduction with references support.
5. Lines 304-305: What is the meaning of "An expert radiologist and radiographer"? Was only one observer involved? Did that observer possess both radiology and radiographer qualifications? Please provide details of the qualification and experience of that observer. Also, what were the image assessment tasks performed by that observer? It is not clearly shown. Did that person just view the images? Did he / she rate the texture, contrast and blur? I could not find the corresponding results presented in the Results section.
6. Line 338: The abbreviation, RIs has been used previously. Why the term, Radiomic indices is shown here? Please be consistent.
7. Line 376: Please fix the typo error, "Therelative".
8. Line 781: Please provide proof for your belief.
9. Lines 870-889: These contents are predominantly descriptions of findings. Where are the discussion points? Also, please cite references.
10. Line 900-926: Please cite references.
11. Only 19 references listed in the References section, not matching the in-text citations.
Author Response
Rev 2:
Dear Reviewer,
We would like to thank the reviewer for their thorough comments on the manuscript. Below is our response to the issues raised in the review.
- Abstract: please provide key figures to support your result description.
We modified the Abstract section to contain more key figures about our findings. For this purpose, direct numeric values and related statements were inserted in the section. However, please accept that for this reason, we should exceed the 200-word limit to do this.
- This manuscript still has 17 illustrations (10 figures and 7 tables) which appear too many.
Thank you very much for this comment; however, we believe that the illustrations in the manuscript are necessary for an easier overview and visualization of our very large number of data. In addition, we tried to find any suggestions for the number of images and tables in the journal instruction section again, but we could not locate related statements. Furthermore, we found numerous articles having more than 17 illustrations in the Diagnostics journal. For example, https://doi.org/10.3390/diagnostics12081952, https://doi.org/10.3390/diagnostics11101875 and https://doi.org/10.3390/diagnostics12081962.
- Content organisation and flow of idea need improvement. For example, the second sentence of the Introduction section should be moved to the beginning. Please review the content organisation and flow of idea for the whole manuscript to improve readability. Some abbreviations, e.g. RI in line 84, etc. are not defined before using. Please address this issue.
According to the reviewer's suggestion, we thoroughly reviewed and modified the manuscript in order to improve its comprehensibility. The phrase “Radiomic indices” was abbreviated as RI in the Abstract; however, we now clarify it in the Introduction.
- The basis of using QR code pattern for printing the phantoms is still not clearly presented in the Introduction. Suggest providing further discussion about this in the Introduction with references support.
Based on the reviewer’s comment, we modified the Introduction to contain the basic idea of the proposed QR code pattern.
- Lines 304-305: What is the meaning of "An expert radiologist and radiographer"? Was only one observer involved? Did that observer possess both radiology and radiographer qualifications? Please provide details of the qualification and experience of that observer. Also, what were the image assessment tasks performed by that observer? It is not clearly shown. Did that person just view the images? Did he / she rate the texture, contrast and blur? I could not find the corresponding results presented in the Results section.
Thank you for this comment; the statements about the observers were not complete and clear. The phrase ‘experts’ in this specific case means 30 and 10 years of clinical experience as a radiologist and radiographer, respectively. They qualitatively assessed which acquisition settings provide the best texture visibility; however, no rating or scoring was performed during this task. The method and results section of the manuscript was modified accordingly.
- Line 338: The abbreviation, RIs has been used previously. Why the term, Radiomic indices is shown here? Please be consistent.
Previously, our first impression was that it was better to start this sentence with the entire form; now, we modified it and started with the abbreviated form.
- Line 376: Please fix the typo error, "Therelative".
Thank you for your comment, we have strived to correct typo errors in the entire manuscript.
- Line 781: Please provide proof for your belief
We omitted the belief word and rewrote the sentence in question. Our proof is mainly lying in our literature-based searches. According to numerous scientific databases, we did not find similar studies and publications.
To search for the following keywords “3D printing”, “radiomics”, “phantom”, and “MRI” in together, Google Scholar provide nearly 800 hits. Applying the same search condition for PubMed, ScienceDirect, and ResearchGate we got 3, 13, and 100 hits, respectively. These obviously overlapped. All our searches were performed in June of 2022.
Most of these studies only mentioned that 3D printing is a possible way to produce radiomics phantoms (10.1016/j.ejmp.2020.02.003). Others have 3D printed phantoms segmented from real patient data (10.1097/RLI.0000000000000530, 10.1016/j.ejmp.2022.04.007, 10.1016/j.urology.2017.08.056) or one with artificially created shapes (10.1002/mp.14173 ).
- Lines 870-889: These contents are predominantly descriptions of findings. Where are the discussion points? Also, please cite references
According to the Reviewer’s comment, we modified this section, adding discussion points to the findings. References have also been included.
- Line 900-926: Please cite references.
We thank the Reviewer; accordingly, references have been inserted into the manuscript.
- Only 19 references listed in the References section, not matching the in-text citations.
We corrected this error in the manuscript and ensured that the Mendeley citation feature worked correctly.

Round 3
Reviewer 2 Report
Thank you for your revision. However, several issues related to my previous comments are still noted as follows.
1. Lines 141-145: Please review the sentence structure and grammar. The sentences look unclear.
2. Lines 265-267: Do the radiologist and radiographer specialise in MRI? If yes, have they got 30 and 10 years of experience in MRI respectively? Please clarify. Also, please clearly define the parameters for indicating best texture visibility.
3. Lines 395-398: Please define what is expert. This is not noted in lines 265-267. Also, the image quality assessment parameters stated there are not defined in lines 265-267.
Author Response
Dear Reviewer,
We thank the Reviewer for his/her helpful comments on the manuscript.
- Lines 141-145: Please review the sentence structure and grammar. The sentences look unclear.
The Reviewer is right to point out that the cited part of the manuscript was misstructured and hard to understand. We have corrected the sentence.
- Lines 265-267: Do the radiologist and radiographer specialise in MRI? If yes, have they got 30 and 10 years of experience in MRI respectively? Please clarify. Also, please clearly define the parameters for indicating best texture visibility.
We thank the Reviewer for their questions. The radiologist (Ervin Berényi) is a full professor with 30 years of MRI experience, and the radiographer (GergÅ‘ Veres) has been working in MR for ten years. No quantitative parameters have been used for evaluating texture visibility; they analyzed the image data qualitatively to determine how close the imaged texture of each object was to the actual pattern and heterogeneity. We modified the section accordingly.
- Lines 395-398: Please define what is expert. This is not noted in lines 265-267. Also, the image quality assessment parameters stated there are not defined in lines 265-267.
According to this comment, and taking into account our answer given in the previous point, we revised the manuscript.
